# Elastomeric Cardiowrap Scaffolds Functionalized with Mesenchymal Stem Cells-Derived Exosomes Induce a Positive Modulation in the Inflammatory and Wound Healing Response of Mesenchymal Stem Cell and Macrophage

**DOI:** 10.3390/biomedicines9070824

**Published:** 2021-07-15

**Authors:** Juan Carlos Chachques, Chiara Gardin, Nermine Lila, Letizia Ferroni, Veronique Migonney, Celine Falentin-Daudre, Federica Zanotti, Martina Trentini, Giulia Brunello, Tiberio Rocca, Vincenzo Gasbarro, Barbara Zavan

**Affiliations:** 1Laboratory of Biosurgical Research (Alain Carpentier Foundation), Pompidu Hospital, University of Paris, 75015 Paris, France; jcchachques@gmail.com (J.C.C.); lilan@gmail.com (N.L.); 2GVM Care & Research, Maria Cecilia Hospital, 48033 Cotignola, Italy; gardin@gvmnet.com (C.G.); ferroni@gvmnet.com (L.F.); 3Department of UMR, University Sorbonne Paris Nord, 93430 Villetaneuse, France; mingonneti@yahoo.com (V.M.); dautreceline@gmail.fr (C.F.-D.); 4Translational Medicine Department, University of Ferrara, 44123 Ferrara, Italy; zanottif@unife.it (F.Z.); trentinimartina@unife.it (M.T.); 5Department of Neurosciences, University of Padova, 35133 Padova, Italy; brunell@unipd.it; 6Division of Internal Medicine, St. Anna Hospital, 44123 Ferrara, Italy; roccat@unife.it (T.R.); gasbarro@unife.it (V.G.); 7Department of Medical Sciences, University of Ferrara, 44123 Ferrara, Italy

**Keywords:** heart failure, myocardial infarction, cardiac tissue engineering, elastomeric scaffold, cardiopatch, cardiowrap, polycaprolactone, mesenchymal stem cells, exosomes

## Abstract

A challenge in contractile restoration of myocardial scars is one of the principal aims in cardiovascular surgery. Recently, a new potent biological tool used within healing processes is represented by exosomes derived from mesenchymal stem cells (MSCs). These cells are the well-known extracellular nanovesicles released from cells to facilitate cell function and communication. In this work, a combination of elastomeric membranes and exosomes was obtained and tested as a bioimplant. Mesenchymal stem cells (MSCs) and macrophages were seeded into the scaffold (polycaprolactone) and filled with exosomes derived from MSCs. Cells were tested for proliferation with an MTT test, and for wound healing properties and macrophage polarization by gene expression. Moreover, morphological analyses of their ability to colonize the scaffolds surfaces have been further evaluated. Results confirm that exosomes were easily entrapped onto the surface of the elastomeric scaffolds, increasing the wound healing properties and collagen type I and vitronectin of the MSC, and improving the M2 phenotype of the macrophages, mainly thanks to the increase in miRNA124 and decrease in miRNA 125. We can conclude that the enrichment of elastomeric scaffolds functionalized with exosomes is as an effective strategy to improve myocardial regeneration.

## 1. Introduction

Cardiovascular diseases (CVD) are the most worldwide cause of mortality and morbidity. Acute myocardial infarction and subsequent heart failure are major health issues as the human adult heart has minimal regenerative capacity [1,2]. The first biological cardiowrap strategy designed to improve systolic contraction and limit ventricular dilatation was the latissimus dorsi dynamic cardiomyoplasty (LD-CMP), which was indicated in severe heart failure patients. This biomechanical surgical approach was developed by Chachques and Carpentoer in France [3]. The LD skeletal-muscle flap was wrapped around the heart and chronically electrostimulated by specific electrodes and implantable pacemaker (cardio-myostimulator). The skeletal muscle contracts in concert with the myocardium during systole improved hemodynamics. More than 2000 patients have been operated worldwide; CMP provides an autologous source of circulatory support [4,5,6]. Tissue engineering is actually considered the most promising approach to investigate the ideal material for clinical use. Scaffolds for cardiovascular regeneration usually consist of a 3D polymeric structure that contributes to cell attachment, organization, proliferation, and interaction of further tissue formation [7,8,9,10]. Several biomaterials have been tested for the fabrication of scaffolds suitable for this purpose, including natural and synthetic biomaterials. The most diffuse strategies are based on synthetic materials, such as poly(ethylene terephtalate) (PET) fabric (DACRON) and expanded polytetrafluoroethylene (ePTFE, e.g., Gore-Tex or IMPRA), which are applied to reconstruct tissue deficiencies in cardiovascular surgery [11,12,13,14]. Despite their wide diffusion they are not clinically ideal due to the lack of novel tissue growth, risks of calcification and infection, lack of biodegradation, and unsuitable mechanical properties. In this view, novel strategies are explored in order to overtake these problems. Recently, Brzeziński et al. [15] proposed a novel holdfast device for a left atrial appendage (LAA) design and 3D printing based on polyamide. This device has been implanted in porcine hearts where, in all cases, the surface of the atrium under the holdfast device had no clots and was smooth with a foreign body reaction comparable to that to control (polyester based) grafts [16]. Among the natural-derived materials, the most used are collagen [17,18,19,20,21,22,23], alginate, a polysaccharide derived from algae [24,25,26], and albumin [27,28]. Promising materials are found in the synthetic group, such as polylactic acid (PLA), which forms lactic after its degradation, polyglycolic acid (PGA), and its copolymer with PLA poly(D,L-lactic-co-glycolic acid) (PLGA), which, even if it is an FDA-approved biomaterial due to their high stiffness, shows a limited capacity to remodel the scaffold and then to complete their maturation, compared to the cardiomyocites [29,30,31,32]. Other strategies are based on elastomeric material, which, thanks to the its properties such as elasticity and strength, could prevent cardiac dilation and preserve cardiac function after myocardial infarction. In this view, great results have been obtained with polylactic (PLA) and poly-ε-caprolactone (PCL) [33,34,35]. Recently, we indeed published our results direct to confirm that electrospun PCL-based fibers are suitable for vascular tissue engineering thanks to its large surface area, flexible surface, and good mechanical characteristics. The enrichment in their elastic component allows for ideal long-term efficacy cardiovascular materials, as our in vivo results confirm [36].

Recently, exosomes have served as substitutes for MSC in cell-free therapy, showing important roles during myocardial infarction injury mediating heart repair [37]. Even if the mechanisms of cardioprotection of MSC-derived exosomes have not been fully elucidated, the modulation of M1/M2 macrophage polarization have gradually attracted researchers [38]. Macrophages are involved in heart injury and repair after MI, whereby macrophages migrate into the ischaemic cardiac tissue and participate in inflammation and cardiac repair. Depletion of macrophages prevents an increase in wound healing and left ventricular remodeling after myocardial injury. Evidence confirms that Exo could improve heart repair via modulation of macrophage phenotype thanks to their ability to promote M2 macrophage and degrade M1 macrophage polarization in an ischaemic heart [38]. Other researchers have showed that MSC-derived exosome treatment could remarkably improve cardiac function by decreasing cardiomyocyte apoptosis. In dilated cardiomyopathy in particular, MSC-derived exosomes could alleviate inflammatory cardiomyopathy and reduce cardiomyocyte apoptosis by regulating the anti-inflammatory macrophage polarization. These support that MSC-Exo can mediate heart repair based on its interaction with macrophages [38]. Even if the specific molecular mechanism of macrophage polarization has not been clearly acknowledged, their potential should not be ignored. Given the emerging therapeutic potential of MSC-derived exosomes in cardiovascular diseases instead of MSCs themselves, the effective bioactive components, mainly represented by miRNA contained in exosomes, play important roles in interaction with recipient cells.

In order to improve their healing activity, mainly due to control of the first inflammatory event, we enriched these fibers with exosomes, i.e., extracellular nanovesicles product from mesenchymal stem cells which have recently gained great attention from researchers thanks to their immune-modulative properties [37,38,39,40]. In vascular remodeling, intercellular communication is a key process. It is based on either direct cell-to-cell contact or paracrine effects. Recently, researchers have focused their attention on extracellular vesicles (EVs), mainly on exosomes [37,38,39,40]. With a size, which is normally between 30–100 nm, exosomes are generated by endosomal pathways and enriched in several specific protein markers such as tetraspanins CD63, CD81, CD9, and CD82, which make them distinguishable from other EVs [41]. It is well known that MSC exosomes improve wound healing by stimulating various cellular signaling pathways [42,43] or affecting the fate decision of some immune cells such as monocytes, thus facilitating a return to immune homeostasis or through the attenuation of excessive inflammation [44,45]. In order to maximize the therapeutic functions of exosomes, the packing of biomaterials with MSC exosomes have to be utilized, confirming better outcomes than naïve treatment in regeneration models. The results derived from the application of supporting materials, such as hydrogels, placenta, or chitosan, confirmed the in vivo stability and retention of the exosome (including their content on proteins and miRNAs), enhancing its therapeutic effectiveness and strengthening the wound repair process [46,47,48]. Recently, our group confirmed that mineral-doped PLA-based porous scaffolds enriched with exosome increased the osteogenic commitment of human adipose mesenchymal stem cells (hAD-MSCs) [49]. In this view, in the present work we used the same strategies to entrap exosome within the fibers of an elastomeric PCL-based scaffolds in order to test how well they enhance healing tissue through the analyses of macrophage polarization and wound healing properties of mesenchymal stem cells.

## 2. Materials and Methods

### 2.1. Elastomeric Scaffolds

PCL electrospun membranes (10 × 10 cm, 1 mm thickness) were produced using medical-grade polycaprolactone PCL Mn = 80,000 g/mol^−1^ from Corbion. Solvents used for the preparation of electrospinning polymer solutions were obtained from Fisher Scientific for chloroform and from Carl Roth for dimethylformamide. Distilled water was obtained from a Millipore Milli-Q Plus water purification system equipped with a 0.22 μm filter (resistivity of 18.2 MΩ cm at 25 °C). Sodium styrene sulfonate (NaSS; Sigma) used for the grafting process was purified by recrystallization in a mixture of water/ethanol (Carlo Erba) (10: 90 *v/v*). The purified NaSS was then dried under atmospheric pressure at 50 °C overnight and then stored at 4 °C [50]. PCL was electrospun using a homemade electrospinning device. PCL was first dissolved in a mixture of chloroform and dimethylformamide (70/30 *v/v*) to prepare a 13% *w*/*v* solution and was stirred overnight before use. The polymer solution was then loaded into a 20 mL glass syringe fitted with a 20 G (0.9 mm) blunt-tipped needle. The solution was continuously ejected using a syringe pump at a rate of 2 mL h^−1^. The voltage used for electrospinning was 9 kV, and the distance between the needle and the collector was 18 cm. The deposition time was 8 h for all experiments. PCL electrospun fiber scaffolds were dried overnight at air pressure and room temperature and placed at 4 °C until further experiments. PCL scaffolds were cut into 3 × 3 × 0.5 cm membranes.

Before the biological assays, grafted PCL fiber scaffold surfaces were washed consecutively under stirring at room temperature with saline aqueous solution 1.5 M sodium chloride (NaCl; Fisher), saline aqueous solution 0.15 M NaCl, and pure water, and with phosphate-buffered saline (PBS) solution (Gibco). Each step lasted 3 h and was repeated three times. The PCL fiber scaffolds were finally air-dried and sterilized by exposure to ultraviolet light for 15 min on each side.

### 2.2. Cells and Exosomes Isolation

Mesenchymal stem cells were bought from Resnova (Rome, Italy). The isolation of PBMCs monocytews was carried out by a Ficoll-Paque gradient method [51]. Briefly, peripheral blood freshly extracted from patients was carefully poured into a tube with 1:4 of Ficoll at the blood/Ficoll, centrifuged at 591× *g* for 30 min at room temperature. The supernatant was discarded. Human adipose mesenchymal stem cells (hAD-MSCs) (Lonza Inc., Walkersville, MD, USA) were bought from Lonza (Lonza Inc., Walkersville, MD, USA) and were cultured with dedicated growth medium (MSCGM™; Bullet Kit, Lonza Inc., Walkersville, MD, USA). Cell cultures were maintained at 37 °C and 5% CO_2_, and the medium was changed twice a week.

To test the commitment of the MSC, cells were detached from plastic cultures or from elastomeric scaffolds enriched with exosomes and cultured into the following differentiation medium:

Osteogenic differentiation medium: NH OsteoDiff Medium (Miltenyi Biotec, Bergish Gladbach, Germany).

Endothelial differentiation medium: DMEM containing 10% FBS plus 0.1 ng/mL human recombinant ECGF, 10 µg/mL human bFGF (Calbiochem, San Diego, CA, USA), and 100 µg/mL porcine heparin (Seromed; Berlin, Germany).

Chondrogenic differentiation medium: NH ChondroDiff Medium (Miltenyi Biotec, Bergish Gladbach, Germany).

Adipogenic differentiation medium: NH AdipoDiff Medium (Miltenyi Biotec, Bergish Gladbach, Germany).

EVs were isolated from the growth medium of MSCs at a density of 10^6^ and at passages 4–6. The cell culture supernatants were collected and centrifuged at 4 °C for 10 min at 200× *g* and then for 10 min at 500× *g*. Culture supernatants were thawed and spun down vertically at 4 °C for 20 min at 2000× *g*, centrifuged horizontally at 100,000× *g* for 75 min. The supernatant was discarded and the pellet containing the exosomes was re-suspended in 1 mL of PBS [49].

### 2.3. Flow Cytometry

As previously described by [52], after dissociation by trypsin, cells were suspended in flow cytometry staining buffer (R&D Systems, Minneapolis, MN, USA) at a final cell concentration of 1 × 10^6^ cells/mL and incubated for 30 min with mouse anti-human CD44 FITC, CD73 APC, CD90 R-PE, CD105 PE-Cy 7, (eBioscience™, Thermo Fisher Scientific, Waltham, MA, USA) in 500 µL of flow cytometry staining buffer. Attune NxT flow cytometer (Thermo Fisher Scientific) was used for the analysis [52].

### 2.4. Characterization of Exosomes Isolated from MSCs

Exosomes were analyzed in size and concentration with a high-resolution system (NanoSight NS300, Malvern Instruments, Malvern, UK) configured with blue 488 nm laser and with a scientific camera (CMOS, Mightex Systems). Each sample from different isolations was recorded three times for 60 s at a 23 °C temperature. Three replicable histograms were created and the values were then averaged [52].

### 2.5. Exosomes Release Profile of the Elastomeric Fibers

The exosome release profile from elastomeric scaffolds was performed by a micro BCA protein assay kit (Beyotime, Haimen, China). Briefly, elastomeric scaffolds containing 1 μg exosomes were placed in the upper transwell chamber, while 100 μL PBS was added in the lower chamber. Next, 10 μL PBS was collected and replaced by 10 μL of fresh PBS on days 0, 3, 6, 9, 12, 15, 18, and 21. The content of released exosomes was then evaluated and expressed as a percentage.

### 2.6. Transmission Electron Microscopy (TEM)

The exosomes were fixed overnight at 4 °C in 2.5% glutaraldehyde/0.1 M sodium cacodylate buffer solution and were subsequently treated with 1% OsO_4_/0.1 M sodium cacodylate buffer. Ethanol solutions at an increased concentration were used for dehydration and then fixed in epoxy resin (EPON™, Hexion, Houston, TX, USA). Ultramicrotome was used for the preparation of ultrathin (LKB, Stockholm, Sweden) which was stained with heavy metal solutions (1% uranyl acetate and 1% lead citrate) and analyzed by TEM (Tecnai G12, FEI Company, Hillsboro, OR, USA) at an acceleration voltage of 100 kV. The image acquisition was performed by a video camera (Tietz, Tietz Video and Image Processing Systems GmbH, Gauting, Germany) and an imaging software (TIA, FEI Company, Hillsboro, OR, USA) [49].

### 2.7. Immunofluorescence

The samples (exosomes and slide) were previously fixed for 10 min in 4% paraformaldehyde (Sigma-Aldrich, St. Louis, MO, USA). Before staining the samples, they were washed 3 times on PBS and incubated for 1 h at room temperature in PBS solution plus 3% bovine serum albumin (BSA, Sigma-Aldrich). After subsequently washing the samples, they were incubated at 4 °C overnight with primary mouse antibodies anti-human CD 81, CD63, (Thermo Fisher Scientific, Waltham, MA, USA) and with fluorescent secondary goat antibodies anti-human (Alexa Fluor 555 dye, Thermo Fisher Scientific). Additional staining was also performed. Next, the exosomes were marked with PKH26 (Red Fluorescent Cell Linker Kits MINI26; Sigma-Aldrich Co., St. Louis, MO, USA) for 5 min at room temperature in a dark room and blocked with fetal bovine serum, according to the manufacturer’s instructions [49].

### 2.8. Scanning Electron Microscopy (SEM)

The SEM analysis was performed at Centro di Analisi e Servizi Per la Certificazione (CEASC, University of Padova, Padova, Italy) (SEM) with a FESEM, QUANTA200, (FEI, Eindhoven, The Netherlands) instrument. The instrument could work at a low voltage (lower than 10 kV) in order to prevent any sample alteration under the electron beam. The samples were dried in the fume hood for 24 h, mounted on metal stubs and sputter-coated with gold palladium, and then analyzed under high vacuum conditions using the secondary electron detector [32].

### 2.9. MTT Assay

The samples were incubated for 3 h at 37 °C in 1 mL of 0.5 mg/mL MTT solution responded in PBS. After removal of the MTT solution, the formazan contented inside the cells was extracted with 0.5 mL of 10% DMSO. For each sample, O.D. values at 570 nm were recorded in duplicate on 200 μL aliquots using a multilabel plate reader (Victor 3, Perkin Elmer, Milano, Italy) [53].

### 2.10. LDH Activity

LDH activity was tested with the LDH Activity Assay Kit (Sigma-Aldrich) on days 3 and 7 of cell culture. Briefly, the culture medium was collected to determine extracellular LDH activity and the intracellular LDH activity was estimated after cells lysis using the assay buffer present in the kit. Both samples were incubated with a supplied reaction mixture, and the final product was measured at 450 nm using Victor 3 plate reader [54]. All conditions were tested in duplicate.

### 2.11. RNA Extraction, First-Strand cDNA Synthesis and Gene Expressione

Total RNA was isolated from cells grown onto scaffolds or onto plastic for 7 days using the total RNA purification Plus kit (Norgen Biotek, Toronto, ON, Canada). The RNA quality and concentration of the samples were measured with the NanoDrop™ ND-1000 (Thermo Fisher Scientific). For each sample, 500 ng of total RNA was reverse-transcribed using an RT2 First Strand kit (Qiagen, Hilden, Germany) in a final reaction volume of 20 μL [39]. Real-time PCR was performed according to the user’s manual on wound healing or inflammatory response and Autoimmunity RT2 profiler PCR Array (Qiagen) or the primer reported on Table 1 with a StepOnePlus™ Real-Time PCR System (Applied Biosystems™, Foster City, CA, USA) using RT2 SYBR Green ROX FAST Master Mix (Qiagen). Thermal cycling and fluorescence detection were as follows: 95 °C for 10 min, followed by 40 cycles of 95 °C for 15 s, and 60 °C for 1 min. At the end of each run, a melting curve analysis was performed using the following program: 95 °C for 1 min, 65 °C for 2 min with optics off, and 65 °C to 95 °C at 2 °C/min with optics on. Each experiment was repeated 3 times, and each measure was repeated 3 times [55].

### 2.12. Statistical Analysis

All results are expressed as the mean ± standard deviation (SD) obtained from at least three independent experiments. Significant differences among groups were determined by analysis of variance (ANOVA), followed by post-hoc Bonferroni tests. The Student’s *t*-test was performed to determine the statistical significance between two samples. Different labels indicate * *p* < 0.05, ** *p* < 0.01, and *** *p* < 0.001.

## 3. Results

### 3.1. Exosomes

Exosomes (Exo) were isolated from MSCs which were previously characterized for their markers. As reported on Figure 1, MSCs were indeed positive for CD44, CD73, CD90. (Figure 1A).

The commitment ability of the MSCs (Figure 1B) was performed by the cultures of the MSCs in the presence of the differentiation culture medium for up to 21 days when we performed gene expression analyses to detect their phenotypes. The genes selected for this screening were as follows:

Endothelial commitment: von Willebrand factor (vWF) and CD31. As reported in B, expression for these markers was detectable whether no collagen type II typical of cartilage was present. Few molecules of ppar gamma and collagen type I were expressed.

Bone commitment: osteopontin, osteonectin, and osteocalcin. Expression of these markers was detectable. Markers such as collagen type II, PPAR-gamma, and adiponectin were completely absent. Low expression of von Willebrand, CD31, and glut 4 was present.

Adipogenic commitment: adiponectin, GLUT4, PPARγ, and collagen type I expression was found. Low expression of CD31 and von Willebrand were present. Collagen type II, osteopontin, osteonectin, and osteocalcin were completely absent.

Exosomes isolated were analyzed for size, concentration, and the presence of CD markers. As reported on Figure 2A, isolated EXO showed an omogeneous diameter from 120 to 160 (average peak 150.8 ± 7.6 nm).

Morphological analyses confirmed that the majority of the vesicles were round, as TEM tests confirmed (Figure 2B). Cytofluorimetric performance revealed the presence on the external membrane of the typical EXO markers, such as CD63 and CD81 (Figure 2C,D).

Elastomeric scaffold was enriched with exosomes and their in vitro releases were evaluated by measuring the presence of exosomes into the cultured medium (complete DMEM) for up to 15 days. As reported in Figure 2E, exosomes were released during the first 5 days, which was faster compared to the following 5 days and when they reached the complete release at day 12.

### 3.2. Elastomeric Fibers Enriched on Exosomes and Their Effect on Cells

Elastomeric scaffolds were enriched with red stained exosomes (Figure 3) in order to evaluate their shape and ability to attach to the fibers. As reported on Figure 3A, EXO were able to maintain their round features. In Figure 3B, SEM analyses confirmed more over their ability to attach to the fibers.

Morphology of macrophages seeded onto elastomerics scaffolds, whether they were enriched with exosomes or not, were evaluated by means of SEM. As reported on Figure 4, cells were able to attach to the fiber, assuming the right features were once activated.

Mitochondrial activity of the MSC seeded onto the scaffolds was tested with the MTT test to evaluate the ability of the scaffolds and to release active exosomes. Results reported on Figure 5 confirmed an increase in time of this value. This means that the cells were able to proliferate. The cytotoxicity of the scaffold enriched with exosomes were eventually evaluated with LDH quantification, given that this test was close to damage of plasma membrane. As reported on Figure 5, the LDH value related to the intracellular presence of the enzyme increased with time. This confirms the good biocompatibility of the scaffold, otherwise its extracellular value would not increase, confirming the absence of any plasma membrane damage.

Biological activity of macrophages and MSC were eventually evaluated by means of gene expression. As seen in Figure 6, mRNA and miRNA expression related to the phenotype acquisition of inflammatory feature M1 or anti-inflammatory feature M2 of the macrophages reported. As it is well evidenced, the presence of exosomes induce a well-defined increase on miRNA expression related to the M2 phenotype (anti-inflammatory), such as miR: 124, 130,483,877,337,546. By contrast, if monocyte are seeded into the elastomeric scaffolds without exosomes, they express all the miRNA and mRNA pattern related to the inflammatory macrophage M1 feature once they are activated. Moreover, activity of macrophages was evaluated in terms of protein secretion. In this case, the levels were reported as a % of increase when the macrophages were seeded onto plastic dishes.

Moreover, wound healing properties of MSC were tested by means of gene expression of all the genes involved on extracellular matrix production, growth factors, integrin, and intracellular signaling. As reported on Figure 7A for gene expression, graphed by the use of a heat map and in terms of evaluation of protein secretion, elastomeric scaffolds always induce an increase on collagen fibers, positively acting on the integrin expression. The presence of exosome shows a positive activity on vitronectine and VEGF.

The commitment ability of the MSC (Figure 8) after detaching from the elastomeric scaffolds enriched with exosomes were not performed by the cultures of the MSC in the presence of a differentiation culture medium up to 21 days when we performed gene expression analyses to detect their phenotypes. The genes selected for this screening were the following:

Endothelial commitment: von Willebrand factor (vWF) and CD31. As reported, expression for these markers was detectable whether no collagen type II typical of cartilage was present. Few molecule of PPAR-gamma and collagen type I were expressed.

Bone commitment: osteopontin, osteonectin, and osteocalcin. Expression of these markers was detectable, there was no presence of markers such as collagen type II, but PPAR-gamma and adiponectin were present. Low expression of von Willebrand, CD31, and glut 4 was also present.

Adipogenic commitment: adiponectin, GLUT4, PPARγ, and collagen type I expression was found. Low expression of CD31 and von Willebrand were present. Collagen type II, osteopontin, osteonectin, and osteocalcin were completely absent.

No differences compared to the MSC properties were present if the cells were cultured in the presence or absence of the exosomes.

## 4. Discussion

Cardiovascular disease (CVD) is still the leading cause of death across the world [56]. It is well known that there is a the limited endogenous regeneration ability of the adult human heart. In the presence of injury, the damage tissue is replaced with a fibrotic scar unable to ensure the contraction, the compliance of viable cardiac muscles, and the essential electronic transduction [57]. To avoid this and to obtain functional tissues, researchers have focused tissue engineering and on novel approaches in cardiac biomedical engineering, including cellular and acellular therapies [58]. This contributes to the development of several scaffolds that are being tested in the preclinical setting and phase I clinical trials as cell delivery vehicles, in order to obtain approved medical device certification. The global cell therapy products market is indeed set to expand and is estimated to attain €7.24 billion by 2025 which would make it the fastest growing branch in the regenerative medicine trade. Several cellular approaches have been proposed, such as functional cardiomyocytes or msenchymal stem cells (MSCs), thanks to their positive influence on antifibrosis, angiogenesis, differentiation of endogenous stem cells, extracellular matrix homeostasis, chemoattraction, and the immunosuppressive effects on promoting inflammatory resolution [59,60,61,62]. Among these, MSCs primarily act on accelerating the transition of macrophage from M1 (that elicits a release of proinflammatory cytokines and exacerbates the ischemic injury) to M2 has been reported to cardiac protective facilitates heart wound healing as something that improves cardiac performance [63,64,65,66,67]. These therapeutic effects are exerted thanks to its paracrine bioactive components, mainly from the exosomes. These extracellular nanovesicles contain the miRNA involved on tissue healing thanks to their immunomodulate properties [68,69,70,71]. Exosomes can easily attach to the biomaterials in order to obtain bioengineered scaffolds for tissue regeneration [72,73,74,75]. This bioengineered acellular therapy represents an optimized method to deliver MSC products that, unlike conventional carriers, offers the benefit of exosomes, which represents the fine dialogue with the immune system and reduced toxic side effects, as well as the ability to penetrate blood vessels, to cross biological barriers (i.e., the blood-brain barrier), and to enter the extracellular matrix. In order to label these advantages, we combined elastomeric scaffolds with exosomes in order to improve the regenerative potential of the final cardio patch. Our results confirm that the exosomes can be absorbed and entrapped inside the fibers, as confirmed by morphological analyses at electron microscopy and fluorescence analyses. The functional biological activity of the exosomes was then evaluated by means of seeding the activated macrophages and mesenchymal stem cells. Our results conduct mainly with gene expression confirmed that macrophages acquire an anti-inflammatory phenotype (M2) thanks to the expression of the gene related to the well-defined miRNA, such as MiR-124, 130,483,877,337,546. Macrophages were identified as the most important cells for the induction tissue regeneration and remodeling in tissue damage, mostly on cardiac tissue. A huge release of damage-associated molecular patterns (DAMPs) from injured cardiac cells activated NF-κB signaling pathway in cardiac macrophages, inducing an increased secretion of inflammatory cytokines (tumor necrosis factor alpha (TNF-α), lymphocyte, and monocyte-recruiting chemokines (CCL-17 and CCL-24), IL-1β), enhancing nitric oxide (NO) production and increasing inducible nitric oxide synthase (iNOS) expression [76]. All these miRNA are known to interact within these events [77] thanks to their ability to down-regulate expression of NF-κB p65 and to induce the reduced production of NO, IL-1β and IL-18 in macrophages colon injury and inflammation [78]. Such an event is well defined since several researchers, such as Yang and colleagues [79], have suggested that modulation of anti-oxidant/oxidant balance was responsible for MSC-EVs-induced effects on macrophage phenotype and function in their excellent works. In line with these findings, MSC-EVs treatment could significantly reduce activation of IL-1 and iNOS-signaling pathways in inflamed phenotype macrophages, with the final attenuation on production of TNF-α, IL-1β, IL-6 and eventually increasing the secretion of IL-10. Exosomes also exert a positive action, increasing the quality and quantity of extracellular matrix compositions which are fundamental for tissue reconstruction, such as collagen fibers and vitronectin. This last protein is known to promote cardiomyocyte adhesion and expression of important cardiomyocyte functions [80,81,82,83].

## 5. Conclusions

In the present work, we tested the possibility to enrich elastic fibers with exosomes with the aim to improve the healing process of the scaffolds once they should be implanted in vivo. Even preliminary and in vitro results confirm that the response of mesenchymal stem cells and macrophages are positive in terms of a morphological and molecular point of view. To note the great ability of the exosome to influence the acquisition of the anti-inflammatory phenotype M2 of the macrophages, we have verified that the fibers are not only able to be enriched with exosomes but are also able to maintain their biological bioactive signals. Recently, there is a growing interest on the application of exosomes in response to the increasing requirement of human health. As natural nano particles, exosomes show several advantages compared to other engineered synthetic nanoparticles thanks to their cell-based biological structures and functions that provide higher chemical stability, their natural biocompatibility, longer distance intercellular communication, and inherent intercellular communication, fusion, and delivery ability. Moreover, several researchers have demonstrated that they are able to target specific tissues and penetrate tight tissue structures, such as the blood–brain barrier. Furthermore, we have to take into account that the production and contents of exosomes are closely related to the cell origin and activity, and, up to now, the lack of criterion for their production may be the main obstacle to clinical translation. In any case, since the study of MSC-exosomes achieved several promising results, we believe that, with further research, all of these problems will be gradually resolved and that we will have a broader space for their application in the future.

## Figures and Tables

**Figure 1 biomedicines-09-00824-f001:**
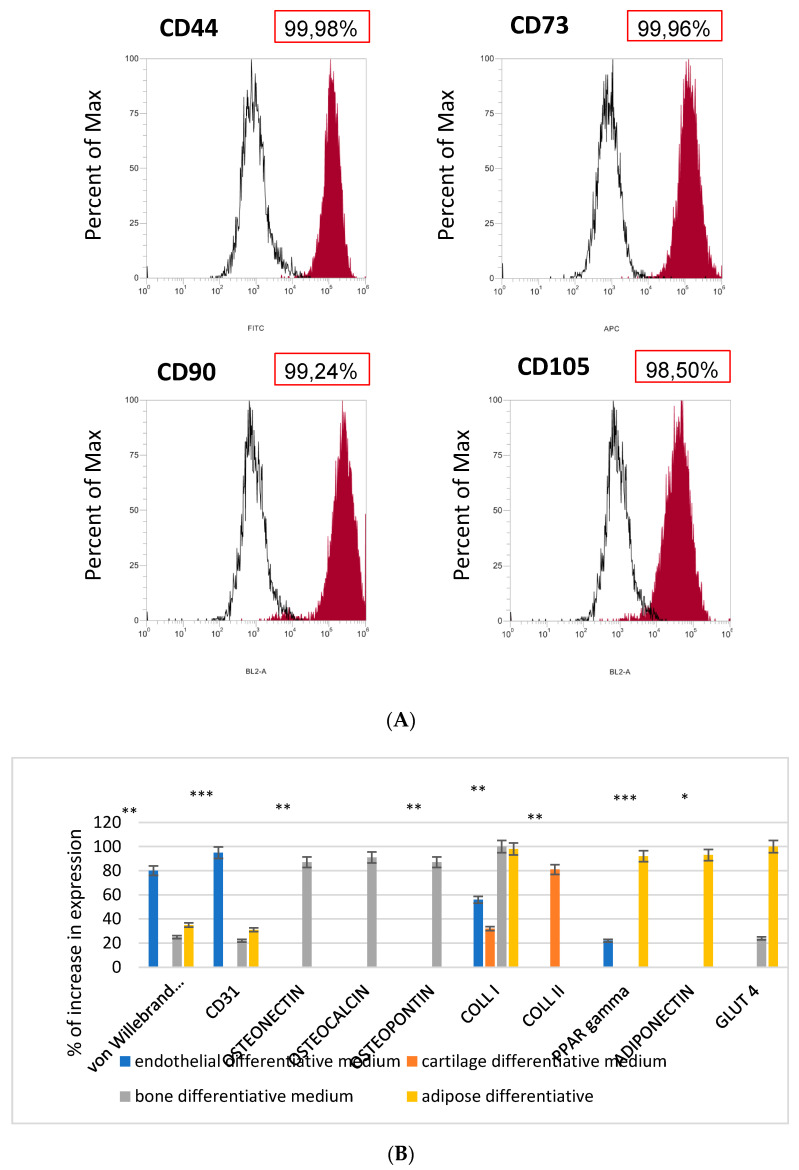
(**A**) Characterization of cell surface markers in MSC by flow cytometry. (A) MSC are positive to CD73, CD44, CD90, and CD105. (**B**) Gene expression according to real-time PCR on MSC in in media that induce differentiation: endothelial, chondrogenic, osteogenic, or adipogenic. One-way analysis of variance (ANOVA) was used for data analyses. Repeated measures ANOVA with a post-hoc analysis using Bonferroni’s multiple comparison. *T*-tests were used to determine significant differences (*p* < 0.05). * *p* < 0.05; ** *p* < 0.01; *** *p* < 0.001. Repeatability was calculated as the standard deviation of the difference between measurements.

**Figure 2 biomedicines-09-00824-f002:**
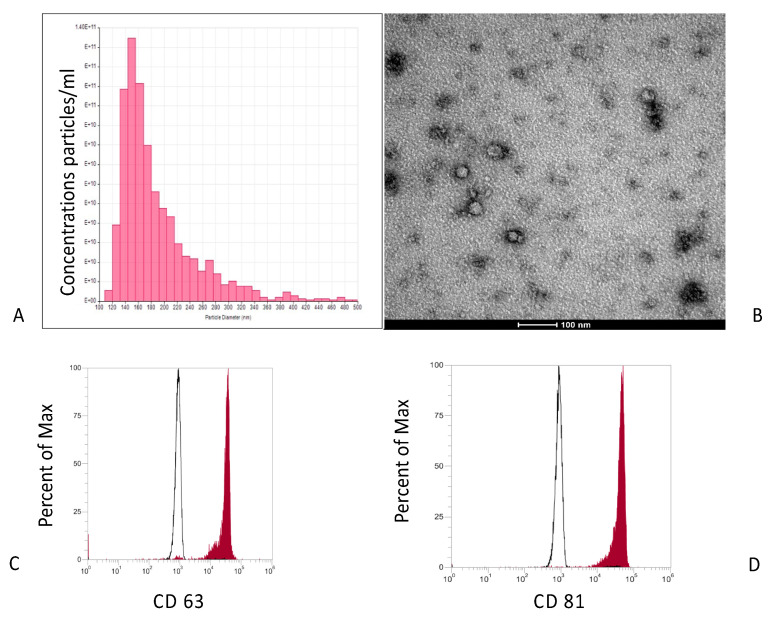
Characterization of exosomes derived from MSCs. (**A**) Size and concentration (by NanoSight analysis) of exosomes secreted by MSCs is reported in the graph. Nanoparticle characterization shows a populations of exosomes with a peak diameter of 100 nm. (**B**–**D**) Immunohistochemistry analysis of the exosome surface markers CD 63 and CD 81 (**D**). Realized profile of exosomes from elastomeric scaffolds (**E**).

**Figure 3 biomedicines-09-00824-f003:**
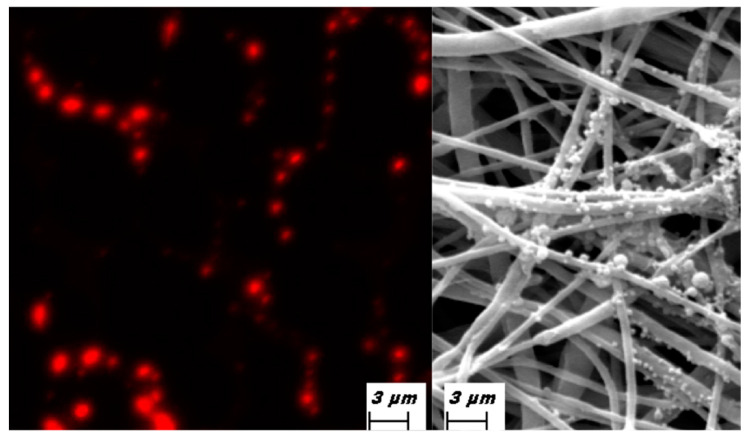
Morphological analyses of of exososme seeded onto cardiowrap fibers by means of immunofluorescence (**A**) were exosomes are in red, or (**B**) Electron microscopy that show the attachmento of exosomes onto the fibers.

**Figure 4 biomedicines-09-00824-f004:**
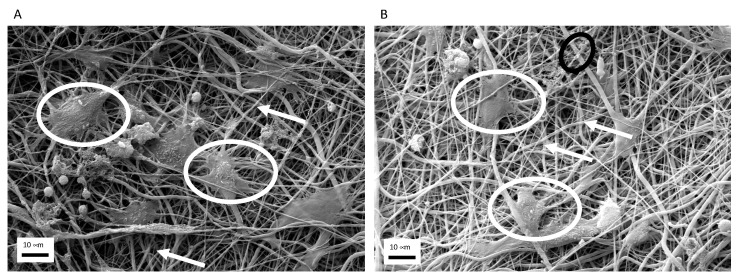
Morphological analyses of of macrophages (white circles) seeded onto elastomeric fibers (white arrows) 200× (**A**) and onto elastomeric fibers enriched with exosomes (**B**) (black circles).

**Figure 5 biomedicines-09-00824-f005:**
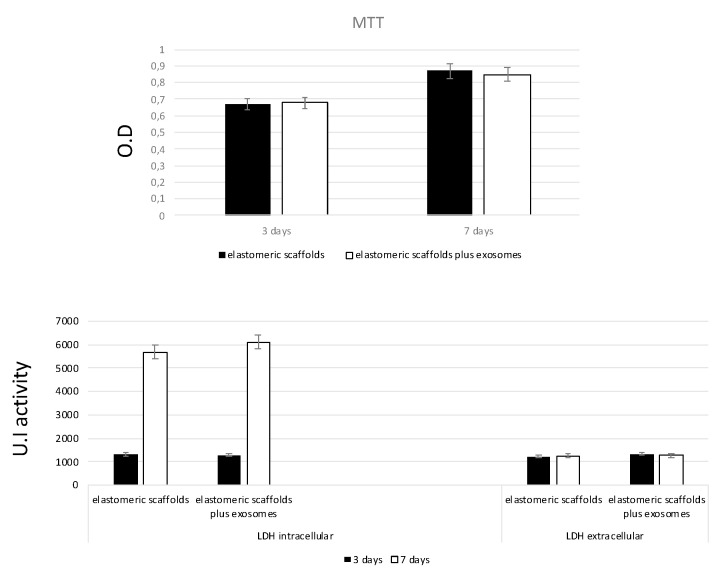
Biocompatibility analyses of scaffolds. MTT elastomeric scaffold for 3 and 7 days. The rate of proliferation MSC increased throughout the culturing period. No significant differences were noted in terms of the proliferation rate among the types of scaffolds at these time points. The toxicity of the surfaces were evaluated by lactate dehydrogenase (LDH) activity assay after 3 and 7 days of culturing. No significant differences were observed among the two types of scaffolds at these time points. Data presented as mean ± standard error (3 measurements) assay of the mesenchymal stem cells (MSCs) cultured onto the elastomeric scaffold.

**Figure 6 biomedicines-09-00824-f006:**
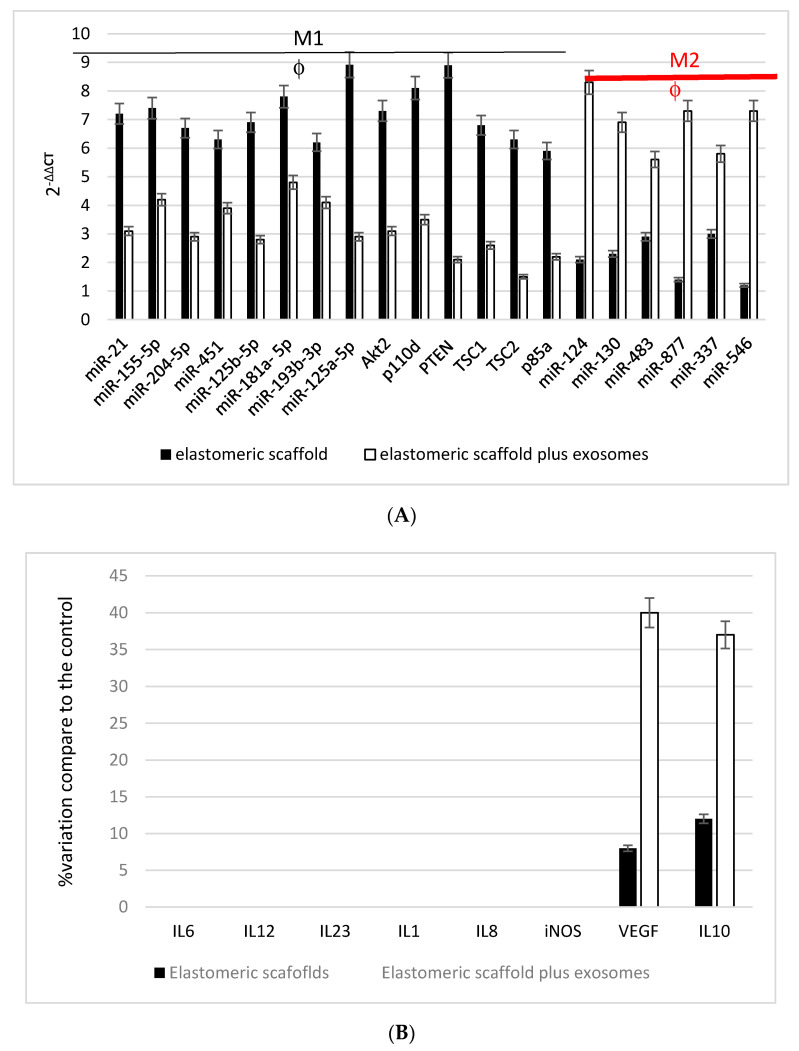
(**A**) semi-quantitative real-time PCR of the mRNA and miRNA levels of macrophages on elastomeric fibers (CTRL) and test conditions (elastomeric fibers plus exosomes) over 2 days of culturing. Data is presented as the mean ± standard error of 3 measurements. *p* ≤ 0.05. (**B**) semi-quantitative evaluation of protein secretion of macrophages seeded on elastomeric scaffolds and onto elastomeric scaffolds plus exosomes.

**Figure 7 biomedicines-09-00824-f007:**
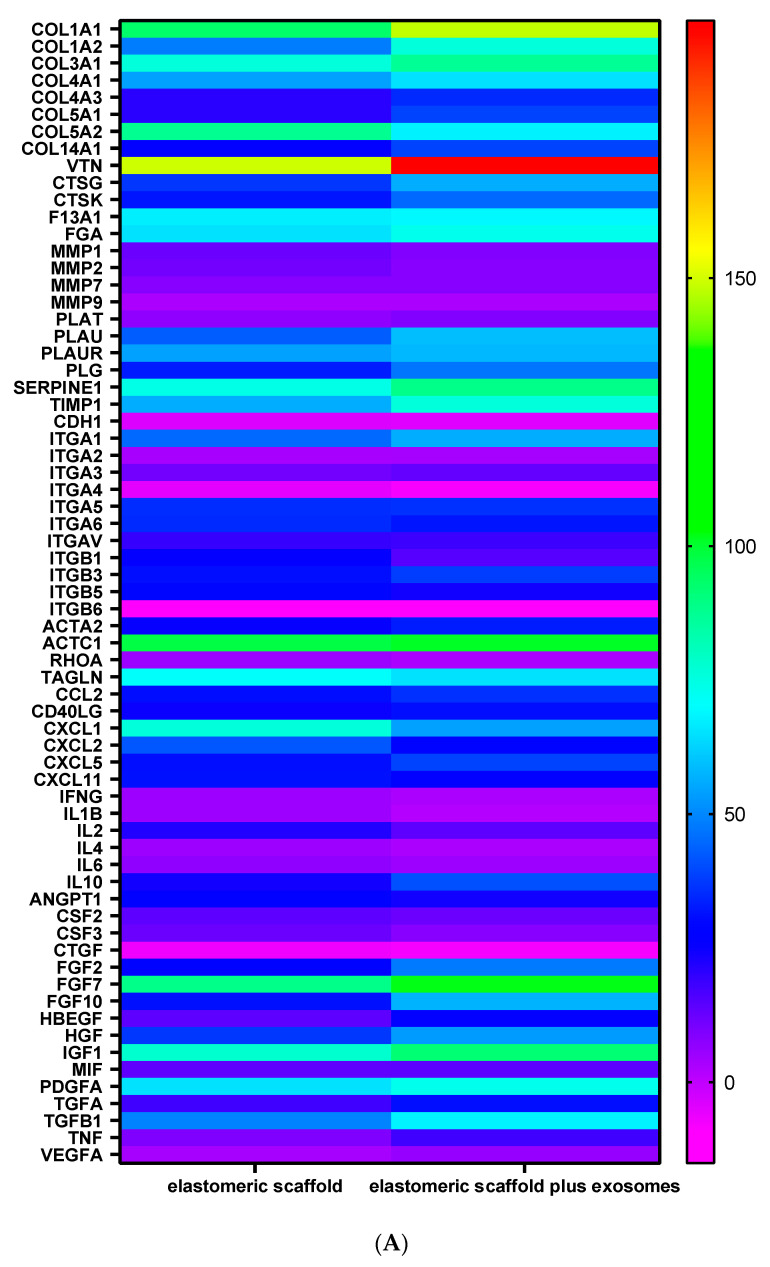
Graphical representation of: (**A**) heat map related to real-time PCR of the mRNA levels of MSC seeded onto elastomeric fibers (CTRL) and test conditions (elastomeric fibers plus exosomes) over 7 days of culturing. (**B**) protein evaluation of secretion product of MSC seeded onto elastomeric scaffolds and onto elastomeric scaffolds plus exosomes. Results are reported as a % of variation compared to the control (MSC cultured onto cell culture dishes surface).

**Figure 8 biomedicines-09-00824-f008:**
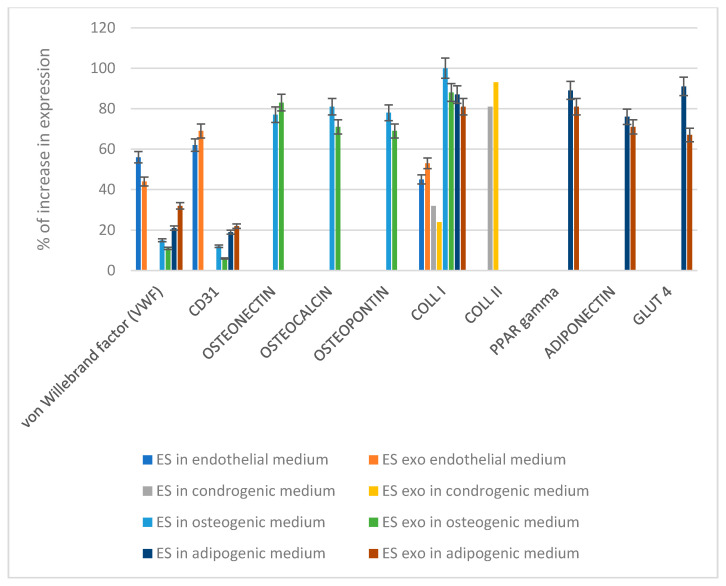
Gene expression according to real-time PCR on MSC previously cultured onto elastomeric scaffolds (ES) or in ES plus exosomes (ES exo) in media that induce differentiation: endothelial, chondrogenic, osteogenic, or adipogenic. Results are reported as the % of increase compared to the control (MSC in no differentiative medium).

**Table 1 biomedicines-09-00824-t001:** Primer sequences.

	FOR	REV	Product (bp)
von Willebrand factor (VWF)	ACGTATGGTCTGTGTGGGATC	GACAAGACACTGCTCCTCCA	159
CD31	TCCAGCCAACTTCACCATCC	TGGGAGAGCATTTCACATACGA	171
OSTEONECTIN	TGCATGTGTCTTAGTCTTAGTCACC	GCTAACTTAGTGCTTACAGGAACCA	186
OSTEOCALCIN	GCAGCGAGGTAGTGAAGAGAC	AGCAGAGCGACACCCTA	193
OSTEOPONTIN	TGGAAAGCGAGGAGTTGAATGG	GCTCATTGCTCTCATCATTGGC	192
COLL I	TGAGCCAGCAGATCGAGA	ACCAGTCTCCATGTTGCAGA	178
COLL II	CAGCAAGAGCAAGGAGAAGAAAC	GTGGTAGGTGATGTTCTGGGA	163
PPAR gamma	CAGGAGATCACAGAGTATGCCAA	TCCCTTGTCATGAAGCCTTGG	149
ADIPONECTIN	GATGAGAGTCCTGGGTGTGAG	CTGGGTAGATATGGGATTCAAGAGA	148
GLUT 4	CCTGATCATTGCGGTCGTG	CCGAGACCAAGGTGAAGACTG	163

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
