# Peer review of "Elastomeric Cardiowrap Scaffolds Functionalized with Mesenchymal Stem Cells-Derived Exosomes Induce a Positive Modulation in the Inflammatory and Wound Healing Response of Mesenchymal Stem Cell and Macrophage"

_biomedicines, 2021, doi:10.3390/biomedicines9070824_

Round 1

Reviewer 1 Report

The study presents the potential cardiac regenerative ability of a scaffold functionalized with MSCs exosomes. The work is intriguing and in line with the latest developments in this area of research. The results are well organized and presented in a logical manner. However, the manuscript has some major issues:

- There are many English grammar and phrasing errors. Some of the identified errors are :

               -lines 40-41: “Acute myocardial infarction and subsequent heart failure are the major health issues this because human adult heart has minimal regenerative capacity.”- delete “this”

               -line 68: “which for lactic after its degradation”, instead of “for”, it should be “forms”

               -line 71, please rephrase “then to ultimately their maturation” and delete the first “s” letter at the beginning of the line

               - line 82-83, please rephrase : “that recently riched great attention from the researcher that to their immune-modulative properties”

               -line 146, please correct “1x106 cells/mL”

               -line 178, please add punctuation “performed Moreover the exosomes”

               - line 185, please correct the letter “n” from “n order”

               -line 226, please correct “concentrationand”

               -lines 230-231, please rephrase “more over cytofluorimetric performance revealed the presence on the external membrane of the EXO markers such as CD63, and CD81”

               -line 233, fig1 legend, please correct the following “MSC are positive 233 to CD73, CD44, CD90, and 105.”

               - line 246 “enriched and not”, I think is “enriched or not”

               - everywhere if you mention miRNAs please provide a “miR-“ before the name of each one, such as, lines 276-277 and line 327 instead of “miRNA: 276124, 130,483,877,337,546.”, there should be “miR-276124, miR-130, miR-483, miR-877, miR-337, miR-546”

               -lines 286-287, replace “and” with “an” and correct the word “increas” – “induce always and increas on collagene fibers”

- please discuss more on the results presented in Figure 7. The overexpression of some genes related to increased wound healing properties of the scaffold, such as increase in Vitronectine, integrines and collagen production should be confirmed through western blotting, for at least 3-4 genes

- focus more in the discussion section on the effects of the presented scaffold on the wound  healing capacity of MSCs and what does it mean cardiac regeneration

Minor issues:

- please provide a more complete definition of the exosomes in the introduction, explain that they also contain mARNs and miRNAs

- The spelling of miRNAs should be uniform in Figure 3

- In Figure 4 legend you mention a black arrow, but in the figure there is a black circle, please correct

Author Response

- There are many English grammar and phrasing errors. Some of the identified errors are :

               -lines 40-41: “Acute myocardial infarction and subsequent heart failure are the major health issues this because human adult heart has minimal regenerative capacity.”- delete “this”

Thanks done

               -line 68: “which for lactic after its degradation”, instead of “for”, it should be “forms”

Thanks done

               -line 71, please rephrase “then to ultimately their maturation” and delete the first “s” letter at the beginning of the line

Thanks done

               - line 82-83, please rephrase : “that recently riched great attention from the researcher that to their immune-modulative properties”

Thanks done

               -line 146, please correct “1x106 cells/mL”

Thanks done

               -line 178, please add punctuation “performed Moreover the exosomes”

Thanks done

               - line 185, please correct the letter “n” from “n order”

Thanks done

               -line 226, please correct “concentrationand”

Thanks done

               -lines 230-231, please rephrase “more over cytofluorimetric performance revealed the presence on the external membrane of the EXO markers such as CD63, and CD81”

Thanks done

               -line 233, fig1 legend, please correct the following “MSC are positive 233 to CD73, CD44, CD90, and 105.”

Thanks done

               - line 246 “enriched and not”, I think is “enriched or not”

Thanks done

               - everywhere if you mention miRNAs please provide a “miR-“ before the name of each one, such as, lines 276-277 and line 327 instead of “miRNA: 276124, 130,483,877,337,546.”, there should be “miR-276124, miR-130, miR-483, miR-877, miR-337, miR-546”

Thanks done

               -lines 286-287, replace “and” with “an” and correct the word “increas” – “induce always and increas on collagene fibers”

Thanks done

- please discuss more on the results presented in Figure 7. The overexpression of some genes related to increased wound healing properties of the scaffold, such as increase in Vitronectine, integrines and collagen production should be confirmed through western blotting, for at least 3-4 genes

Thanks we added Elisa test for vitronectin, collagene and other secreted protein production

- focus more in the discussion section on the effects of the presented scaffold on the wound  healing capacity of MSCs and what does it mean cardiac regeneration

Thanks done

Minor issues:

- please provide a more complete definition of the exosomes in the introduction, explain that they also contain mARNs and miRNAs

Thanks done

- The spelling of miRNAs should be uniform in Figure 3

Thanks done

- In Figure 4 legend you mention a black arrow, but in the figure there is a black circle, please correct

Thanks done

Reviewer 2 Report

Article: Elastomeric cardiowrap scaffolds functionalized with Mesenchymal stem cells derived Exosomes induce a positive modulation in the inflammatory and wound healing response of Mesenchymal stem cell and macrophage

Manuscript ID: biomedicines-1227433

  1. Article Summary

In this primary article Chachques and colleagues reports the development of an elastomeric scaffold capable of binding cells and exosomes. In their work they demonstrate seeding the scaffold with mesenchymal stem cells (MSCs), MSC derived exosomes, and macrophages. After seeding for 7 days the group shows a decrease in immune markers in the macrophages. They subsequently conclude that their scaffold is an effective method of delivering exosomes to macrophages with potential implications as a repair scaffold following cardiac injury. While the data is very early and largely proof-of-concept, the ability to selectively target MSC exosome delivery is a benefit to the field.

  1. Comment 1:

The authors note that, in total, they seeded exosomes, MSCs and macrophages onto their scaffold. Some of the experimental conditions are not entirely clear to me. Figure 6 is, to me, the most important figure in the paper as it shows a clear change in the macrophage phenotype. However, were these macrophages cultured on the scaffold with exosomes alone, or with exosomes and MSCs? Please clarify.

  1. Comment 2:

The work is interesting, and the need for local delivery of MSC-derived exosomes is certainly an active and exciting field. However, an important limitation in this work is that the group is showing the effect of direct seeding of macrophages and exosomes together on the scaffold. This is not the same as what would occur in clinical environment, where the scaffold seeded with exosomes would presumably be added to the surface of the injured myocardium. As a result, the exosome signaling would need to traverse the scaffold and epicardium before potentially interacting with macrophages. While the work demonstrated here is early and proof of principal, it would be worthwhile to see if the same effect on the immune signature is seen if the exosome-seeded scaffold is co-cultured with macrophages in a tissue culture system (for example, the Transwell System by Corning) rather than directly seeded together on the scaffold as this would better mimic the use of the scaffold in vivo.

III. Comment 3:

I think the true advantage to the system the authors have developed is that the scaffold stabilizes the MSCs and prevents them from flowing away from the site of injury (versus if the MSCs or exosomes were injected directly). There is no comment on this in the manuscript. Do they authors have an idea of how long their exosomes are stable on their scaffold? How does this compare to an isolated exosome in culture without the scaffold?

  1. Comment 4

I am curious if the authors have seeded their scaffold with MSCs directly, rather than the exosomes, and if a similar effect on the macrophage is noted. Exosomes themselves deliver their cargo then are destroyed, but if the authors can deliver an MSC, which is constantly making exosomes, they may see a stronger effect.

  1. Comment 5

The authors do an excellent job characterizing the surface markers of the MSCs they use to culture their exosomes. Typically, MSC identity requires demonstration of cell surface markers as well as differentiation capacity. Have the authors checked the differentiation capacity of their MSCs – that is the ability to form adipocytes, chondrocytes and osteocytes?

  1. Comment 6

I have some difficulty following their RT-PCR data. Is this one culture experiment, or does “elastomeric scaffold” and “elastomeric scaffold plus exosomes” represent more than one experiment per condition that was then pooled? Please clarify.

VII. Comment 7

I am not sure Fig 7 adds much to the paper, and the findings are difficult to interpret. None of the genes seem to have a particularly large change in expression level. Does the scale represent intensity? In which case are these averages or one experiment? What is the fold difference in expression? Perhaps most importantly, it does not seem likely that growing MSCs in the scaffold with exosomes would have any difference from growing MSCs alone as the MSCs presumably are making and releasing their own exosomes. If the authors wish to show that the MSCs maintain their regenerative capacity on the scaffold, isolating the cells off the scaffold after 7 days and re-checking surface markers and differentiation capacity and their ability to make exosomes would be more thorough way of demonstrating this.

VIII. Comment VIII

As a corollary to the data in Fig 6, I would suggest the authors collect conditioned media from the macrophage and exosome samples and compare this to macrophages cultured in the scaffold alone. There are several commercially available protein microarrays that look at cytokines and it might be interesting to see if there is a difference in the cytokines released in the media from the macrophages on the scaffold with or without the exosomes. This would give you a functional read out rather than just the expression data.

  1. English

There are several grammatical, typological, and spelling errors throughout the manuscript that are too numerous to list that do interfere with the reading of the manuscript. Please have the manuscript edited for this.

Author Response

The authors note that, in total, they seeded exosomes, MSCs and macrophages onto their scaffold. Some of the experimental conditions are not entirely clear to me. Figure 6 is, to me, the most important figure in the paper as it shows a clear change in the macrophage phenotype. However, were these macrophages cultured on the scaffold with exosomes alone, or with exosomes and MSCs? Please clarify.

 Thanks we explained that macrophages have been cultured with only exosomes, more over we added an explanation of the advantage of the enrichment with only exosomes compare the one with MSC

  1. Comment 2:

The work is interesting, and the need for local delivery of MSC-derived exosomes is certainly an active and exciting field. However, an important limitation in this work is that the group is showing the effect of direct seeding of macrophages and exosomes together on the scaffold. This is not the same as what would occur in clinical environment, where the scaffold seeded with exosomes would presumably be added to the surface of the injured myocardium. As a result, the exosome signaling would need to traverse the scaffold and epicardium before potentially interacting with macrophages. While the work demonstrated here is early and proof of principal, it would be worthwhile to see if the same effect on the immune signature is seen if the exosome-seeded scaffold is co-cultured with macrophages in a tissue culture system (for example, the Transwell System by Corning) rather than directly seeded together on the scaffold as this would better mimic the use of the scaffold in vivo.

Thanks done, but the aim of this article in not demonstrate the activity of the exosome but the possibility to use a scaffold for their delivery

III. Comment 3:

I think the true advantage to the system the authors have developed is that the scaffold stabilizes the MSCs and prevents them from flowing away from the site of injury (versus if the MSCs or exosomes were injected directly). There is no comment on this in the manuscript. Do they authors have an idea of how long their exosomes are stable on their scaffold? How does this compare to an isolated exosome in culture without the scaffold?

Thanks we added a test to confirm their in vitro delivery

  1. Comment 4

I am curious if the authors have seeded their scaffold with MSCs directly, rather than the exosomes, and if a similar effect on the macrophage is noted. Exosomes themselves deliver their cargo then are destroyed, but if the authors can deliver an MSC, which is constantly making exosomes, they may see a stronger effect.

Thanks but aim of the paper is not to demonstrate the validity of the exosome as tool for regeneration but the production of a free cell based therapy. For this we don t want compare the effect of a scaffold enriched with the exosome with a scaffolds enriched with MSC

  1. Comment 5

The authors do an excellent job characterizing the surface markers of the MSCs they use to culture their exosomes. Typically, MSC identity requires demonstration of cell surface markers as well as differentiation capacity. Have the authors checked the differentiation capacity of their MSCs – that is the ability to form adipocytes, chondrocytes and osteocytes?

 Thanks done, we added the gene expression related markers

  1. Comment 6

I have some difficulty following their RT-PCR data. Is this one culture experiment, or does “elastomeric scaffold” and “elastomeric scaffold plus exosomes” represent more than one experiment per condition that was then pooled? Please clarify.

Thanks the data represent more then one experiment (3 in particular) per condition. We added this information

VII. Comment 7

I am not sure Fig 7 adds much to the paper, and the findings are difficult to interpret. None of the genes seem to have a particularly large change in expression level. Does the scale represent intensity? In which case are these averages or one experiment? What is the fold difference in expression? Perhaps most importantly, it does not seem likely that growing MSCs in the scaffold with exosomes would have any difference from growing MSCs alone as the MSCs presumably are making and releasing their own exosomes. If the authors wish to show that the MSCs maintain their regenerative capacity on the scaffold, isolating the cells off the scaffold after 7 days and re-checking surface markers and differentiation capacity and their ability to make exosomes would be more thorough way of demonstrating this.

 Thanks to the referee, more over:

-Fig 7 adds much to the paper, and the findings are difficult to interpret. None of the genes seem to have a particularly large change in expression level. Does the scale represent intensity? In which case are these averages or one experiment? What is the fold difference in expression?

Fig. 7 is an heat map that represent the most commont method for visualising and interpreting gene expression data can be used for both microarray and RNA-seq experiments. The heatmap may also be combined with clustering methods which group genes and/or samples together based on the similarity of their gene expression pattern. This can be useful for identifying genes that are commonly regulated, or biological signatures associated with a particular condition (e.g presence of absence of exosomes). In heat maps the data is displayed in a grid where each row represents a gene and each column represents a sample. The colour and intensity of the boxes is used to represent changes (not absolute values) of gene expression. In the example below, red represents up-regulated genes and blue represents down-regulated genes. Black represents unchanged expression.

-Perhaps most importantly, it does not seem likely that growing MSCs in the scaffold with exosomes would have any difference from growing MSCs alone as the MSCs presumably are making and releasing their own exosomes. If the authors wish to show that the MSCs maintain their regenerative capacity on the scaffold, isolating the cells off the scaffold after 7 days and re-checking surface markers and differentiation capacity and their ability to make exosomes would be more thorough way of demonstrating this.

We follow the suggestion of the referee, confirming  that the MSC maintained their differentiation capacity, more over during the in vitro coltures and during the detachment phases the surfaces markers were damaged so it was impossible to perform their surface marker present by citofluorimetry

VIII. Comment VIII

As a corollary to the data in Fig 6, I would suggest the authors collect conditioned media from the macrophage and exosome samples and compare this to macrophages cultured in the scaffold alone. There are several commercially available protein microarrays that look at cytokines and it might be interesting to see if there is a difference in the cytokines released in the media from the macrophages on the scaffold with or without the exosomes. This would give you a functional read out rather than just the expression data.

Thanks done
